# Ulam–Hyers Stability and Uniqueness for Nonlinear Sequential Fractional Differential Equations Involving Integral Boundary Conditions

**Areen Al-khateeb** [1,*] **, Hamzeh Zureigat** [1] **, Osama Ala'yed** [1] **and Sameer Bawaneh** [2]

1   Department of Mathematics, Faculty of Science and Information Technology, Jadara University, Irbid 21110, Jordan; hamzeh.zu@jadara.edu.jo (H.Z.); alayedo@jadara.edu.jo (O.A.)
2   Department of Software Engineering, Faculty of Architecture and Engineering, Rauf Denktas University, Nicosia 139149, North Cyprus; Sameer.bawaneh@rdu.edu.tr
*   Correspondence: areen.k@jadara.edu.jo; Tel.: +962-779-474-718

**Abstract:** Fractional-order boundary value problems are used to model certain phenomena in chemistry, physics, biology, and engineering. However, some of these models do not meet the existence and uniqueness required in the mainstream of mathematical processes. Therefore, in this paper, the existence, stability, and uniqueness for the solution of the coupled system of the Caputo-type sequential fractional differential equation, involving integral boundary conditions, was discussed, and investigated. Leray–Schauder's alternative was applied to derive the existence of the solution, while Banach's contraction principle was used to examine the uniqueness of the solution. Moreover, Ulam–Hyers stability of the presented system was investigated. It was found that the theoretical-related aspects (existence, uniqueness, and stability) that were examined for the governing system were satisfactory. Finally, an example was given to illustrate and examine certain related aspects.

**Keywords:** sequential fractional differential equation; fixed-point theorem; Ulam–Hyers stability; fractional differential equation





## 1. Introduction

In recent decades, the field of fractional-order boundary value problems has been discussed by several scientific researchers across the world. This is evident from the number of significant studies of fractional-order boundary value problems that mainly focus on extending and transforming such problems from the theoretical to the application aspect, in order to make them applicable for certain real-life phenomena. The fractional calculus essentially involves differentiation and integration to an arbitrary order, which is considered as an important tools that have facilitated many real-life phenomena in considerable fields such as physics, biology, and chemistry (see [1–4]). In addition, engineering is considered to be one of the main fields that benefits from fractional calculus, due to providing a full and comprehensive description of some complex engineering models.

Moreover, the significance of fractional calculus stretches further than scientific areas, to several other areas that influence human civilization in general. As an outcome of these efforts, many practical mathematical models that are expressed based on fractional differential equations have been developed, providing infinite description support of such mathematical models and developing a novel strategy for use in other practical fields. This leads to a new path of research that aims to have more collaboration between mathematicians and other researchers. In addition, actual practicality is considered to be one of the essential advantages of fractional-order models, such as those mentioned in the following articles [5–18]. Recently, Boutiara et al. [19] discussed the solution of a nonlinear sequential q-difference equation based on the Caputo fractional quantum derivatives, with

nonlocal boundary value conditions containing Riemann–Liouville fractional quantum integrals at four points. The criteria and conditions of the existence and uniqueness of the solutions to the given Caputo fractional q-difference boundary value problem have been derived in this study. The stability of the proposed equation was investigated based on Ulam–Hyers stability, and the results obtained were examined by providing two examples.

The coupled system of the Caputo-type sequential fractional differential is considered as one of the most important tools available to model and simulate certain real-life phenomena. Therefore, there is a need to investigate the related theoretical aspects. Motivated by the above discussion and our review of the literature, this paper aims to discuss and analyze the following coupled system of Caputo-type sequential fractional differential equations. In particular, we aimed to investigate the existence, stability, and uniqueness of the solution to the coupled system of Caputo-type sequential fractional differential equations:

$$
\begin{cases}
{}^{c}D^{\alpha-1}(D+k)x(t) = f(t, x(t), y(t)), \ t \in [0, T], \ 1 < \alpha \leq 2, \ k > 0, \\
{}^{c}D^{\beta-1}(D+k)y(t) = g(t, x(t), y(t)), \ t \in [0, T], \ 1 < \beta \leq 2, \ k < 0,
\end{cases}
\tag{1}
$$

supplemented with integral boundary conditions of the form:

$$
\begin{cases}
\displaystyle\int_{0}^{T} x(s)ds = \rho_1 y(\zeta_1), \ \int_{0}^{T} x'(s)ds = \rho_2 y'(\zeta_2), \\
\displaystyle\int_{0}^{T} y(s)ds = \mu_1 x(\eta_1), \ \int_{0}^{T} y'(s)ds = \mu_2 x'(\eta_2), \ \eta_1, \ \eta_2, \zeta_1, \zeta_2 \in [0, T],
\end{cases}
\tag{2}
$$

where ${}^{c}D^{k}$ denote the Caputo fractional derivatives of order $k$, $k = \alpha, \beta$, and $f, g : [0, T] \times \mathbb{R}^2 \to \mathbb{R}$, are given continuous functions, and $\rho_1, \rho_2, \mu_1, \mu_2$ are real constants.

## 2. Preliminaries

Firstly, we recall the definitions of fractional derivatives and integrals [1,2].

**Definition 1.** *The Caputo derivative of fractional order $\gamma$ for $(k-1)-$ times absolutely continuous function $g : [0, \infty) \to \mathbb{R}$ is defined as:*

$$
{}^{c}D^{\gamma}g(s) = \frac{1}{\Gamma(k-\gamma)} \int_{0}^{s} (s-t)^{k-\gamma-1} g^{(m)}(t)dt, \ k-1 < \gamma < k, \ k = [\gamma] + 1,
$$

*where $[\gamma]$ is the integer part of the real number $\gamma$.*

**Definition 2.** *The Riemann–Liouville fractional integral of order $\gamma$ for a continuous function $g$ is given by:*

$$
I^{\gamma}g(t) = \frac{1}{\Gamma(\gamma)} \int_{0}^{t} \frac{g(s)}{(t-s)^{1-\gamma}}dt, \ \gamma > 0,
$$

*provided that the right-hand side is point-wise defined on $[0, \infty)$.*

**Definition 3.** *Due to Miller–Ross [3], the sequential fractional derivative for a sufficiently smooth function $g(t)$ is defined as:*

$$
D^{m}g(t) = D^{m_1}D^{m_2}\dots D^{m_n}g(t),
$$

*where $m = (m_1, m_2, \dots, m_n)$ is a multi-index.*

We prove the following auxiliary lemma to find the solution for the problems (1) and (2).

**Lemma 1.** *Let,* $\phi \in C([0,T],\mathbb{R})$. *Then the unique solution of the problem:*

$$
\begin{cases}
{}^{C}D^{\alpha-1}(D+k)x(t) = \psi(t), 1 < \alpha \le 2, \\
{}^{C}D^{\beta-1}(D+k)y(t) = \phi(t), 1 < \beta \le 2, \\
\int_0^T x(s)ds = \rho_1 y(\zeta_1), \int_0^T x'(s)ds = \rho_2 y'(\zeta_2) \\
\int_0^T y(s)ds = \mu_1 x(\eta_1), \int_0^T y'(s)ds = \mu_2 x'(\eta_2), k > 0, t \in [0,T],
\end{cases}
\tag{3}
$$

*is:*

$$
\begin{aligned}
x(t) = \Delta e^{-kt} &\left[ \mu_2 (I^{\alpha-1}\psi)(\eta_2) - k\mu_2 \int_0^{\eta_2} e^{-k(\eta_2-s)} (I^{\alpha-1}\psi)(s)ds \right. \\
&\left. - \int_0^T (I^{\beta-1}\phi)(s)ds + k \int_0^T \int_0^x e^{-k(x-s)} (I^{\beta-1}\phi)(s)dsdx \right] \\
+ \lambda e^{-kt} &\left[ \rho_2 (I^{\beta-1}\phi)(\zeta_2) - k\rho_2 \int_0^{\zeta_2} e^{-k(\zeta_2-s)} (I^{\beta-1}\phi)(s)ds - \int_0^T (I^{\alpha-1}\psi)(s)ds \right. \\
&\left. + k \int_0^T \int_0^x e^{-k(x-s)} (I^{\alpha-1}\psi)(s)dsdx \right] \\
+ \frac{1}{T^2 - \mu_1\rho_1} &\left[ \frac{A\mu_2}{k} (I^{\alpha-1}\psi)(\eta_2) - A\mu_2 \int_0^{\eta_2} e^{-k(\eta_2-s)} (I^{\alpha-1}\psi)(s)ds \right. \\
&- \frac{A}{k} \int_0^T (I^{\beta-1}\phi)(s)ds + (A-\rho_1) \int_0^T \int_0^x e^{-k(x-s)} (I^{\beta-1}\phi)(s)dsdx \\
&+ \frac{B\rho_2}{k} (I^{\beta-1}\phi)(\zeta_2) - B\rho_2 \int_0^{\zeta_2} e^{-k(\zeta_2-s)} (I^{\beta-1}\phi)(s)ds - \frac{B}{k} \int_0^T (I^{\alpha-1}\psi)(s)ds \\
&+ (B-T) \int_0^T \int_0^x e^{-k(x-s)} (I^{\alpha-1}\psi)(s)dsdx + T\rho_1 \int_0^{\zeta_1} e^{-k(\zeta_1-s)} (I^{\beta-1}\phi)(s)ds \\
&\left. + \mu_1\rho_1 \int_0^{\eta_1} e^{-k(\eta_1-s)} (I^{\alpha-1}\psi)(s)ds \right] + \int_0^t e^{-k(t-s)} (I^{\alpha-1}\psi)(s)ds,
\end{aligned}
\tag{4}
$$

*and:*

$$
\begin{aligned}
y(t) = \theta e^{-kt} &\left[ \mu_2 (I^{\alpha-1}\psi)(\eta_2) - k\mu_2 \int_0^{\eta_2} e^{-k(\eta_2-s)} (I^{\alpha-1}\psi)(s)ds \right. \\
&\left. - \int_0^T (I^{\beta-1}\phi)(s)ds + k \int_0^T \int_0^x e^{-k(x-s)} (I^{\beta-1}\phi)(s)dsdx \right] \\
+ \tau e^{-kt} &\left[ \rho_2 (I^{\beta-1}\phi)(\zeta_2) - k\rho_2 \int_0^{\zeta_2} e^{-k(\zeta_2-s)} (I^{\beta-1}\phi)(s)ds - \int_0^T (I^{\alpha-1}\psi)(s)ds \right. \\
&\left. + k \int_0^T \int_0^x e^{-k(x-s)} (I^{\alpha-1}\psi)(s)dsdx \right] \\
+ \frac{1}{\omega} &\left[ \frac{C\mu_2}{k} (I^{\alpha-1}\psi)(\eta_2) - C\mu_2 \int_0^{\eta_2} e^{-k(\eta_2-s)} (I^{\alpha-1}\psi)(s)ds - \frac{C}{k} \int_0^T (I^{\beta-1}\phi)(s)ds \right. \\
&+ (C-T) \int_0^T \int_0^x e^{-k(x-s)} (I^{\beta-1}\phi)(s)dsdx + \frac{D\rho_2}{k} (I^{\beta-1}\phi)(\zeta_2) \\
&- D\rho_2 \int_0^{\zeta_2} e^{-k(\zeta_2-s)} (I^{\beta-1}\phi)(s)ds - \frac{D}{k} \int_0^T (I^{\alpha-1}\psi)(s)ds \\
&+ (D-\mu_1) \int_0^T \int_0^x e^{-k(x-s)} (I^{\alpha-1}\psi)(s)dsdx + \mu_1\rho_1 \int_0^{\zeta_1} e^{-k(\zeta_1-s)} (I^{\beta-1}\phi)(s)ds \\
&\left. + T\mu_1 \int_0^{\eta_1} e^{-k(\eta_1-s)} (I^{\alpha-1}\psi)(s)ds \right] + \int_0^t e^{-k(t-s)} (I^{\beta-1}\phi)(s)ds,
\end{aligned}
\tag{5}
$$

*where:*

$$\omega = T^2 - \mu_1\rho_1 \neq 0, \ \Delta = \frac{-k\rho_2 e^{-k\zeta_2}}{M}, \ \lambda = \frac{e^{-kT}-1}{M}, \ \sigma = \Delta k\mu_2 e^{-k\eta_2} - 1$$

$$M = \left(e^{-kT} - 1\right)^2 - k^2\mu_2\rho_2 e^{-k(\zeta_2+\eta_2)} \neq 0, \ \theta = \frac{-\sigma}{e^{-kT}-1}, \ \tau = \frac{\lambda}{e^{-kT}-1},$$

$$A = \left[\theta kT\rho_1 e^{-k\zeta_1} + k\mu_1\rho_1\Delta e^{-k\eta_1} + \left(e^{-kT}-1\right)(\Delta T + \theta\rho_1)\right],$$

$$B = \left[\tau kT\rho_1 e^{-k\zeta_1} + k\mu_1\rho_1\lambda e^{-k\eta_1} + \left(e^{-kT}-1\right)(\lambda T + \tau\rho_1)\right],$$

$$C = \left[\Delta kT\mu_1 e^{-k\eta_1} + k\mu_1\rho_1\theta e^{-k\zeta_1} + \left(e^{-kT}-1\right)(\theta T + \Delta\mu_1)\right],$$

$$D = \left[\lambda kT\mu_1 e^{-k\eta_1} + k\mu_1\rho_1\tau e^{-k\zeta_1} + \left(e^{-kT}-1\right)(\tau T + \lambda\mu_1)\right].$$

**Proof.** The general solutions of the sequential fractional differential equations [20–23] in (3) are known as:

$$x(t) = a_0 e^{-kt} + a_1 + \int_0^t e^{-k(t-s)}\left(I^{\alpha-1}\psi\right)(s)ds, \tag{6}$$

$$y(t) = b_0 e^{-kt} + b_1 + \int_0^t e^{-k(t-s)}\left(I^{\beta-1}\phi\right)(s)ds, \tag{7}$$

observe:

$$lx'(t) = -ka_0 e^{-kt} + \left(I^{\alpha-1}\psi\right)(t) - k\int_0^t e^{-k(t-s)}\left(I^{\alpha-1}\psi\right)(s)ds,$$

$$y'(t) = -kb_0 e^{-kt} + \left(I^{\beta-1}\phi\right)(t) - k\int_0^t e^{-k(t-s)}\left(I^{\beta-1}\phi\right)(s)ds,$$

where $a_i, b_i \in \mathbb{R}, i = 0, 1$ are arbitrary constants.

Applying the conditions:

$$\int_0^T x'(s)ds = \rho_2 y'(\zeta_2), \ \int_0^T y'^{(s)}ds = \mu_2 x'(\eta_2).$$

Then we obtain:

$$a_0 = \Delta\left[\mu_2\left(I^{\alpha-1}\psi\right)(\eta_2) - k\mu_2\int_0^{\eta_2} e^{-k(\eta_2-s)}\left(I^{\alpha-1}\psi\right)(s)ds\right.$$

$$-\int_0^T \left(I^{\beta-1}\phi\right)(s)ds + k\int_0^T\int_0^x e^{-k(x-s)}\left(I^{\beta-1}\phi\right)(s)dsdx\right]$$

$$+\lambda\left[\rho_2\left(I^{\beta-1}\phi\right)(\zeta_2) - k\rho_2\int_0^{\zeta_2} e^{-k(\zeta_2-s)}\left(I^{\beta-1}\phi\right)(s)ds - \int_0^T\left(I^{\alpha-1}\psi\right)(s)ds\right.$$

$$+k\int_0^T\int_0^x e^{-k(x-s)}\left(I^{\alpha-1}\psi\right)(s)dsdx\right],$$

and:

$$b_0 = \theta\left[\mu_2\left(I^{\alpha-1}\psi\right)(\eta_2) - k\mu_2\int_0^{\eta_2} e^{-k(\eta_2-s)}\left(I^{\alpha-1}\psi\right)(s)ds\right.$$

$$-\int_0^T \left(I^{\beta-1}\phi\right)(s)ds + k\int_0^T\int_0^x e^{-k(x-s)}\left(I^{\beta-1}\phi\right)(s)dsdx\right]$$

$$+\tau\left[\rho_2\left(I^{\beta-1}\phi\right)(\zeta_2) - k\rho_2\int_0^{\zeta_2} e^{-k(\zeta_2-s)}\left(I^{\beta-1}\phi\right)(s)ds - \int_0^T\left(I^{\alpha-1}\psi\right)(s)ds\right.$$

$$+k\int_0^T\int_0^x e^{-k(x-s)}\left(I^{\alpha-1}\psi\right)(s)dsdx\right]$$

In view of the conditions $\int_0^T x(s)ds = \rho_1 y(\zeta_1)$, $\int_0^T y(s)ds = \mu_1 x(\eta_1)$, we get:

$$
\begin{aligned}
a_1 = \frac{1}{\omega}\Bigg[ & \frac{A\mu_2}{k}\left(I^{\alpha-1}\psi\right)(\eta_2) - A\mu_2 \int_0^{\eta_2} e^{-k(\eta_2-s)}\left(I^{\alpha-1}\psi\right)(s)ds - \frac{A}{k}\int_0^T \left(I^{\beta-1}\phi\right)(s)ds \\
& + (A-\rho_1)\int_0^T\int_0^x e^{-k(x-s)}\left(I^{\beta-1}\phi\right)(s)dsdx + \frac{B\rho_2}{k}\left(I^{\beta-1}\phi\right)(\zeta_2) \\
& - B\rho_2\int_0^{\zeta_2} e^{-k(\zeta_2-s)}\left(I^{\beta-1}\phi\right)(s)ds - \frac{B}{k}\int_0^T\left(I^{\alpha-1}\psi\right)(s)ds \\
& + (B-T)\int_0^T\int_0^x e^{-k(x-s)}\left(I^{\alpha-1}\psi\right)(s)dsdx + T\rho_1\int_0^{\zeta_1} e^{-k(\zeta_1-s)}\left(I^{\beta-1}\phi\right)(s)ds \\
& + \mu_1\rho_1\int_0^{\eta_1} e^{-k(\eta_1-s)}\left(I^{\alpha-1}\psi\right)(s)ds \Bigg],
\end{aligned}
$$

and:

$$
\begin{aligned}
b_1 = \frac{1}{\omega}\Bigg[ & \frac{C\mu_2}{k}\left(I^{\alpha-1}\psi\right)(\eta_2) - C\mu_2 \int_0^{\eta_2} e^{-k(\eta_2-s)}\left(I^{\alpha-1}\psi\right)(s)ds - \frac{C}{k}\int_0^T \left(I^{\beta-1}\phi\right)(s)ds \\
& + (C-T)\int_0^T\int_0^x e^{-k(x-s)}\left(I^{\beta-1}\phi\right)(s)dsdx + \frac{D\rho_2}{k}\left(I^{\beta-1}\phi\right)(\zeta_2) \\
& - D\rho_2\int_0^{\zeta_2} e^{-k(\zeta_2-s)}\left(I^{\beta-1}\phi\right)(s)ds - \frac{D}{k}\int_0^T\left(I^{\alpha-1}\psi\right)(s)ds \\
& + (D-\mu_1)\int_0^T\int_0^x e^{-k(x-s)}\left(I^{\alpha-1}\psi\right)(s)dsdx + \mu_1\rho_1\int_0^{\zeta_1} e^{-k(\zeta_1-s)}\left(I^{\beta-1}\phi\right)(s)ds \\
& + T\mu_1\int_0^{\eta_1} e^{-k(\eta_1-s)}\left(I^{\alpha-1}\psi\right)(s)ds \Bigg]
\end{aligned}
$$

Substituting the values of $a_0, a_1, b_0, b_1$ in $(6), (7)$ we obtain $(4)$ and $(5)$, which completes the proof. □

### 3. Existence Results

Let the space $Q = \{x(t)|x(t) \in C[0,T]\}$, endowed with the norm $\|x\| = max\{|x(t)|, t \in [0,T]\}$. It is clear that $(Q, \|.\|)$ is a Banach space. Moreover, let $S = \{y(t)|y(t) \in C[0,T]\}$, endowed with the norm $\|y\| = max\{|y(t)|, t \in [0,T]\}$. The product space $(Q \times S, \|(x,y)\|)$ is also a Banach space with the norm $\|(x,y)\| = \|x\| + \|y\|$.

In view of Lemma 1 we define the operator $Z : Q \times S \to Q \times S$ by:

$$Z(x,y)(t) = (Z_1(x,y)(t), Z_2(x,y)(t)),$$

where:

$$
\begin{aligned}
Z_1(x,y)(t) =& \Delta e^{-kt}\left[\mu_2\left(I^{\alpha-1}f\right)(\eta_2) - k\mu_2\int_0^{\eta_2} e^{-k(\eta_2-s)}\left(I^{\alpha-1}f\right)(s)ds\right.\\
& \left. -\int_0^T \left(I^{\beta-1}g\right)(s)ds + k\int_0^T \int_0^x e^{-k(x-s)}\left(I^{\beta-1}g\right)(s)dsdx\right]\\
& +\lambda e^{-kt}\left[\rho_2\left(I^{\beta-1}g\right)(\zeta_2) - k\rho_2\int_0^{\zeta_2} e^{-k(\zeta_2-s)}\left(I^{\beta-1}g\right)(s)ds - \int_0^T \left(I^{\alpha-1}f\right)(s)ds\right.\\
& \left. +k\int_0^T \int_0^x e^{-k(x-s)}\left(I^{\alpha-1}f\right)(s)dsdx\right]\\
& +\frac{1}{\omega}\left[\frac{A\mu_2}{k}\left(I^{\alpha-1}f\right)(\eta_2) - A\mu_2\int_0^{\eta_2} e^{-k(\eta_2-s)}\left(I^{\alpha-1}f\right)(s)ds - \frac{A}{k}\int_0^T \left(I^{\beta-1}g\right)(s)ds\right.\\
& +(A-\rho_1)\int_0^T \int_0^x e^{-k(x-s)}\left(I^{\beta-1}g\right)(s)dsdx + \frac{B\rho_2}{k}\left(I^{\beta-1}g\right)(\zeta_2)\\
& -B\rho_2\int_0^{\zeta_2} e^{-k(\zeta_2-s)}\left(I^{\beta-1}g\right)(s)ds - \frac{B}{k}\int_0^T \left(I^{\alpha-1}f\right)(s)ds\\
& +(B-T)\int_0^T \int_0^x e^{-k(x-s)}\left(I^{\alpha-1}f\right)(s)dsdx + T\rho_1\int_0^{\zeta_1} e^{-k(\zeta_1-s)}\left(I^{\beta-1}g\right)(s)ds\\
& \left. +\mu_1\rho_1\int_0^{\eta_1} e^{-k(\eta_1-s)}\left(I^{\alpha-1}f\right)(s)ds\right] + \int_0^t e^{-k(t-s)}\left(I^{\alpha-1}f\right)(s)ds,
\end{aligned}
\tag{8}
$$

and:

$$
\begin{aligned}
Z_2(x,y)(t) =& \theta e^{-kt}\left[\mu_2\left(I^{\alpha-1}f\right)(\eta_2) - k\mu_2\int_0^{\eta_2} e^{-k(\eta_2-s)}\left(I^{\alpha-1}f\right)(s)ds\right.\\
& \left. -\int_0^T \left(I^{\beta-1}g\right)(s)ds + k\int_0^T \int_0^x e^{-k(x-s)}\left(I^{\beta-1}g\right)(s)dsdx\right]\\
& +\tau e^{-kt}\left[\rho_2\left(I^{\beta-1}g\right)(\zeta_2) - k\rho_2\int_0^{\zeta_2} e^{-k(\zeta_2-s)}\left(I^{\beta-1}g\right)(s)ds - \int_0^T \left(I^{\alpha-1}f\right)(s)ds\right.\\
& \left. +k\int_0^T \int_0^x e^{-k(x-s)}\left(I^{\alpha-1}f\right)(s)dsdx\right]\\
& +\frac{1}{\omega}\left[\frac{C\mu_2}{k}\left(I^{\alpha-1}f\right)(\eta_2) - C\mu_2\int_0^{\eta_2} e^{-k(\eta_2-s)}\left(I^{\alpha-1}f\right)(s)ds - \frac{C}{k}\int_0^T \left(I^{\beta-1}g\right)(s)ds\right.\\
& +(C-T)\int_0^T \int_0^x e^{-k(x-s)}\left(I^{\beta-1}g\right)(s)dsdx + \frac{D\rho_2}{k}\left(I^{\beta-1}g\right)(\zeta_2)\\
& -D\rho_2\int_0^{\zeta_2} e^{-k(\zeta_2-s)}\left(I^{\beta-1}g\right)(s)ds - \frac{D}{k}\int_0^T \left(I^{\alpha-1}f\right)(s)ds\\
& +(D-\mu_1)\int_0^T \int_0^x e^{-k(x-s)}\left(I^{\alpha-1}f\right)(s)dsdx + \mu_1\rho_1\int_0^{\zeta_1} e^{-k(\zeta_1-s)}\left(I^{\beta-1}g\right)(s)ds\\
& \left. +T\mu_1\int_0^{\eta_1} e^{-k(\eta_1-s)}\left(I^{\alpha-1}f\right)(s)ds\right] + \int_0^t e^{-k(t-s)}\left(I^{\beta-1}g\right)(s)ds.
\end{aligned}
\tag{9}
$$

**Theorem 1.** *Assume $f, g : C([0,T] \times \mathbb{R}^2 \to \mathbb{R}$ are jointly continuous functions, and there exists constants $h_1, h_2 \in \mathbb{R}$, such that $\forall\, x_1, x_2, y_1, y_2 \in \mathbb{R}, \forall t \in [0,T]$ we have:*

$$
\begin{aligned}
|f(t,x_1,x_2) - f(t,y_1,y_2)| &\le h_1(|x_2 - x_1| + |y_2 - y_1|),\\
|g(t,x_1,x_2) - f(t,y_1,y_2)| &\le h_2(|x_2 - x_1| + |y_2 - y_1|).
\end{aligned}
$$

*If:*

$$h_1(M_1 + M_3) + h_2(M_2 + M_4) < 1,$$

*then the boundary value problems (1) and (2) have a unique solution on $[0, T]$, where:*

$$M_1 = \left[ \frac{(\alpha+1)|\Delta\mu_2|e^{-kT}\eta_2{}^{\alpha}\left(\alpha\eta_2{}^{-1}+k\right)+|\lambda|e^{-kT}T^{\alpha}(\alpha+1+kT)}{\Gamma(\alpha+2)} \right] + \frac{T^{\alpha}}{\Gamma(\alpha+1)} +$$
$$\left[ \frac{|A\mu_2|\eta_2{}^{\alpha}(\alpha+1)\left(\alpha\eta_2{}^{-1}+k\right)+T^{\alpha}(|B|(\alpha+1)+kT|B-T|)+k(\alpha+1)|\mu_1\rho_1|\eta_1{}^{\alpha}}{k\Gamma(\alpha+2)|T^2-\mu_1\rho_1|} \right],$$

$$M_2 = \left[ \frac{(\beta+1)T^{\beta}\left(|\Delta|e^{-kT}+kT\right)+|\lambda\rho_2|e^{-kT}\zeta_2{}^{\beta}(\beta+1)\left(\beta\zeta_2{}^{-1}+k\right)}{\Gamma(\beta+2)} \right] +$$
$$\left[ \frac{T^{\beta-1}\left(|A|\left(\beta^2+\beta\right)+k|A-\rho_1|T^2\right)+(\beta+1)|B\rho_2|\zeta_2{}^{\beta}\left(\beta\zeta_2{}^{-1}+k\right)+k|T\rho_1|\zeta_1{}^{\beta}}{k\Gamma(\beta+2)|T^2-\mu_1\rho_1|} \right],$$

$$M_3 = \left[ \frac{(\alpha+1)|\theta\mu_2|e^{-kT}\eta_2{}^{\alpha}\left(\alpha\eta_2{}^{-1}+k\right)+|\tau|e^{-kT}T^{\alpha}(\alpha+1+kT)}{\Gamma(\alpha+2)} \right] +$$
$$\left[ \frac{|C\mu_2|\eta_2{}^{\alpha}(\alpha+1)\left(\alpha\eta_2{}^{-1}+k\right)+T^{\alpha}(|D|(\alpha+1)+kT|D-\mu_1|)+k(\alpha+1)|\mu_1 T|\eta_1{}^{\alpha}}{k\Gamma(\alpha+2)|T^2-\mu_1\rho_1|} \right],$$

$$M_4 = \left[ \frac{(\beta+1)T^{\beta}\left(|\theta|e^{-kT}+kT\right)+|\tau\rho_2|e^{-kT}\zeta_2{}^{\beta}(\beta+1)\left(\beta\zeta_2{}^{-1}+k\right)}{\Gamma(\beta+2)} \right] + \frac{T^{\beta}}{\Gamma(\beta+1)} +$$
$$\left[ \frac{T^{\beta-1}\left(|C|\left(\beta^2+\beta\right)+k|C-T|T^2\right)+(\beta+1)|D\rho_2|\zeta_2{}^{\beta}\left(\beta\zeta_2{}^{-1}+k\right)+k|\mu_1\rho_1|\zeta_1{}^{\beta}}{k\Gamma(\beta+2)|T^2-\mu_1\rho_1|} \right].$$

**Proof.** Define $\displaystyle \sup_{0 \le t \le T} f(t,0,0) = f_0 < \infty$, $\displaystyle \sup_{0 \le t \le T} g(t,0,0) = g_0 < \infty$ and $\Omega_{\varepsilon} = \{(x,y) \in Q \times S : (x,y) \le \varepsilon\}$ and $\varepsilon > 0$ such that:

$$\varepsilon \ge \frac{(M_1 + M_3)f_0 + (M_2 + M_4)g_0}{1 - [h_1(M_1 + M_3) + h_2(M_2 + M_4)]}.$$

Firstly, show that $Z\Omega_{\varepsilon} \subseteq \Omega_{\varepsilon}$.

By our assumption, for $(x,y) \in \Omega_{\varepsilon}, t \in [0, T]$ , we have:

$$|f(t,x(t),y(t))| \le |f(t,x(t),y(t)) - f(t,0,0)| + |f(t,0,0)|,$$

$$\le h_1(|x(t)| + |y(t)|) + f_0 \le h_1(x+y) + f_0,$$

$$\le h_1\varepsilon + f_0, \tag{10}$$

and:

$$|g(t,x(t),y(t))| \le h_2(|x(t)| + |y(t)|) + g_0 \le h_2(x+y) + g_0,$$

$$\le h_2\varepsilon + g_0, \tag{11}$$

which leads to:

$$|Z_1(x,y)(t)| \leq |\Delta| e^{-kT} \Big[ |\mu_2| (I^{\alpha-1}|f|)(\eta_2) + k|\mu_2| \int_0^{\eta_2} e^{-k(\eta_2-s)} (I^{\alpha-1}|f|)(s)ds$$

$$+ \int_0^T (I^{\beta-1}|g|)(s)ds + k \int_0^T \int_0^x e^{-k(x-s)} (I^{\beta-1}|g|)(s)dsdx \Big]$$

$$+ |\lambda| e^{-kT} \Big[ |\rho_2| (I^{\beta-1}|g|)(\zeta_2) + k|\rho_2| \int_0^{\zeta_2} e^{-k(\zeta_2-s)} (I^{\beta-1}|g|)(s)ds + \int_0^T (I^{\alpha-1}|f|)(s)ds$$

$$+ k \int_0^T \int_0^x e^{-k(x-s)} (I^{\alpha-1}|f|)(s)dsdx \Big]$$

$$+ \frac{1}{|\omega|} \Big[ \frac{|A\mu_2|}{k} (I^{\alpha-1}|f|)(\eta_2) + |A\mu_2| \int_0^{\eta_2} e^{-k(\eta_2-s)} (I^{\alpha-1}|f|)(s)ds + \frac{|A|}{k} \int_0^T (I^{\beta-1}|g|)(s)ds$$

$$+ |A - \rho_1| \int_0^T \int_0^x e^{-k(x-s)} (I^{\beta-1}|g|)(s)dsdx + \frac{|B\rho_2|}{k} (I^{\beta-1}|g|)(\zeta_2)$$

$$+ |B\rho_2| \int_0^{\zeta_2} e^{-k(\zeta_2-s)} (I^{\beta-1}|g|)(s)ds + \frac{|B|}{k} \int_0^T (I^{\alpha-1}|f|)(s)ds$$

$$+ |B - T| \int_0^T \int_0^x e^{-k(x-s)} (I^{\alpha-1}|f|)(s)dsdx$$

$$+ T|\rho_1| \int_0^{\zeta_1} e^{-k(\zeta_1-s)} (I^{\beta-1}|g|)(s)ds + |\mu_1\rho_1| \int_0^{\eta_2} e^{-k(\eta_2-s)} (I^{\alpha-1}|f|)(s)ds \Big]$$

$$+ 0 \leq \overset{sup}{t} \leq T \int_0^t e^{-k(t-s)} (I^{\alpha-1}|f|)(s)ds.$$

Using (10) and (11) to get:

$$|Z_1(x,y)(t)| \leq \Big[ |\Delta| e^{-kT} \mu_2 \Big( (I^{\alpha-1}1)(\eta_2) + k \int_0^{\eta_2} e^{-k(\eta_2-s)} (I^{\alpha-1}1)(s)ds \Big)$$

$$+ |\lambda| e^{-kT} \Big( \int_0^T (I^{\alpha-1}1)(s)ds + k \int_0^T \int_0^x e^{-k(x-s)} (I^{\alpha-1}1)(s)dsdx \Big)$$

$$+ \frac{1}{|\omega|} \Big( \frac{|A\mu_2|}{k} (I^{\alpha-1}1)(\eta_2) + |A\mu_2| \int_0^{\eta_2} e^{-k(\eta_2-s)} (I^{\alpha-1}1)(s)ds + \frac{|B|}{k} \int_0^T (I^{\alpha-1}1)(s)ds$$

$$+ |B - T| \int_0^T \int_0^x e^{-k(x-s)} (I^{\alpha-1}1)(s)dsdx + |\mu_1\rho_1| \int_0^{\eta_1} e^{-k(\eta_1-s)} (I^{\alpha-1}1)(s)ds \Big)$$

$$+ \int_0^T e^{-k(T-s)} (I^{\alpha-1}1)(s)ds \Big] ||f||$$

$$+ \Big[ |\Delta| e^{-kT} \Big( \int_0^T (I^{\beta-1}1)(s)ds + k \int_0^T \int_0^x e^{-k(x-s)} (I^{\beta-1}1)(s)dsdx \Big)$$

$$+ |\lambda| e^{-kT} \Big( |\rho_2| (I^{\beta-1}1)(\zeta_2) + k|\rho_2| \int_0^{\zeta_2} e^{-k(\zeta_2-s)} (I^{\beta-1}1)(s)ds \Big)$$

$$+ \frac{1}{|\omega|} \Big( \frac{|A|}{k} \int_0^T (I^{\beta-1}1)(s)ds + |A - \rho_1| \int_0^T \int_0^x e^{-k(x-s)} (I^{\beta-1}1)(s)dsdx + \frac{|B\rho_2|}{k} (I^{\beta-1}1)(\zeta_2)$$

$$+ |B\rho_2| \int_0^{\zeta_2} e^{-k(\zeta_2-s)} (I^{\beta-1}1)(s)ds \Big) \Big] ||g||$$

$$\leq \left[ \frac{(\alpha+1)|\Delta\mu_2|e^{-kT}\eta_2{}^\alpha(\alpha\eta_2{}^{-1}+k)+|\lambda|e^{-kT}T^\alpha(\alpha+1+kT)}{\Gamma(\alpha+2)} + \frac{T^\alpha}{\Gamma(\alpha+1)} \right.$$

$$\left. + \frac{|A\mu_2|\eta_2{}^\alpha(\alpha+1)(\alpha\eta_2{}^{-1}+k)+T^\alpha(|B|(\alpha+1)+kT|B-T|)+k(\alpha+1)|\mu_1\rho_1|\eta_1{}^\alpha}{k\Gamma(\alpha+2)|\omega|} \right] ||f||$$

$$+ \left[ \frac{(\beta+1)T^\beta(|\Delta|e^{-kT}+kT)+|\lambda\rho_2|e^{-kT}\zeta_2{}^\beta(\beta+1)(\beta\zeta_2{}^{-1}+k)}{\Gamma(\beta+2)} \right.$$

$$\left. + \frac{T^{\beta-1}(|A|(\beta^2+\beta)+k|A-\rho_1|T^2)+(\beta+1)|B\rho_2|\zeta_2{}^\beta(\beta\zeta_2{}^{-1}+k)+k|T\rho_1|\zeta_1{}^\beta}{k\Gamma(\alpha+2)|\omega|} \right] ||g||$$

Hence, by (8) we have:

$$\|Z_1(x,y)\| \leq (h_1M_1+h_2M_2)\varepsilon + (M_1f_0+M_2g_0) \leq \frac{\varepsilon}{2}. \tag{12}$$

In similar way, we get:

$$\|Z_2(x,y)\| \leq (h_1M_3+h_2M_4)r + (M_3f_0+M_4g_0) \leq \frac{\varepsilon}{2}. \tag{13}$$

From (12) and (13), we obtain:

$$\|Z(x,y)\| \leq \varepsilon.$$

Now, show that $Z$ is a contraction.

Let $(x_1,y_1),(x_2,y_2) \in Q \times S, \forall t \in [0,T]$, then we get:

$$\|Z_1(x_1,y_1)-Z_1(x_2,y_2)\| \leq h_1M_1(\|x_1-x_2\|+\|y_1-y_2\|)+h_2M_2(\|x_1-x_2\|+\|y_1-y_2\|), \tag{14}$$

$$\|Z_2(x_1,y_1)-Z_2(x_2,y_2)\| \leq h_1M_3(\|x_1-x_2\|+\|y_1-y_2\|)+h_2M_4(\|x_1-x_2\|+\|y_1-y_2\|). \tag{15}$$

From (14) and (15), we deduced that:

$$\|Z(x_1,y_1)-Z(x_2,y_2)\| \leq (h_1(M_1+M_3)+h_2(M_2+M_4))(\|x_1-x_2\|+\|y_1-y_2\|).$$

Since $h_1(M_1+M_3)+h_2(M_2+M_4) < 1$, therefore, $Z$ is a contraction operator. Thus, by Banach's fixed point theorem, the operator $Z$ has a unique fixed point on $[0,T]$, which is the unique solution of the problem (1) and (2), and completes the proof. $\square$

The second result is based on the Leray–Schauder alternative.

**Lemma 2** (Leray–Schauder alternative [18], p. 4). *Let $F : E \to E$ be a completely continuous operator (i.e., a map restricted to any bounded set in $E$ is compact). Let $E(F) = \{x \in E : x = \lambda F(x) \text{ for some } 0 < \lambda < 1\}$. Then, either the set $E(F)$ is unbounded, or $F$ has at least one fixed point.*

**Theorem 2.** *Assume $f, g : C([0,T] \times \mathbb{R}^2 \to \mathbb{R}$ are continuous functions and there exists positive real constants $\theta_i, \vartheta_i(i = 0, 1, 2)$ such that $\forall x_i \in \mathbb{R}, (i = 1, 2)$ we have:*

$$|f(t,x_1,x_2)| \leq \theta_0+\theta_1|x_1|+\theta_2|x_2|,$$
$$|g(t,x_1,x_2)| \leq \vartheta_0+\vartheta_1|x_1|+\vartheta_2|x_2|.$$

*If:*

$$(M_1+M_3)\theta_1+(M_2+M_4)\vartheta_1 < 1,$$

*and:*

$$(M_1+M_3)\theta_2+(M_2+M_4)\vartheta_2 < 1.$$

*Then the problems (1) and (2) have at least one solution.*

**Proof.** The proof will be divided into several steps [24–26].

**Step 1.** Show that $Z$ is completely continuous. The continuity of the operator holds true because of the continuity of the function $f, g$.

Let R be a bounded set in $\Omega_\varepsilon = \{(x, y) \in Q \times S : \|(x, y)\| \leq \varepsilon\}$. Then, there exists positive constants $n_1, n_2$ such that:

$$|f(t, x(t), y(t))| \leq n_1, \ |g(t, x(t), y(t))| \leq n_2, \ \forall t \in [0, T],$$

then, for any $(x, y) \in R$ we have $|Z_1(x, y)(t)| \leq M_1 n_1 + M_2 n_2$, which implies that:

$$\|Z_1(x, y)\| \leq M_1 n_1 + M_2 n_2.$$

Similarly, we get $\|Z_2(x, y)\| \leq M_3 n_1 + M_4 n_2$.

Thus, it follows from the above inequalities that the operator $Z$ is uniformly bounded, since:

$$\|Z(x, y)\| \leq (M_1 + M_3) n_1 + (M_2 + M_4) n_2.$$

Next, we show that the operator is equicontinuous.

Let $t_1, t_2 \in [0, T]$ with $t_1 < t_2$. Then we have:

$$|Z_1(x, y)(t_2) - Z_1(x, y)(t_1)|$$

$$\leq |\Delta| e^{-k(t_2 - t_1)} \left[ |\mu_2| \left( I^{\alpha-1} |f| \right) (\eta_2) + k|\mu_2| \int_0^{\eta_2} e^{-k(\eta_2 - s)} \left( I^{\alpha-1} |f| \right)(s) ds \right.$$

$$+ \int_0^T \left( I^{\beta-1} |g| \right)(s) ds + k \int_0^T \int_0^x e^{-k(x-s)} \left( I^{\beta-1} |g| \right)(s) ds dx \Bigg]$$

$$+ |\lambda| e^{-k(t_2 - t_1)} \left[ |\rho_2| \left( I^{\beta-1} |g| \right) (\zeta_2) + k|\rho_2| \int_0^{\zeta_2} e^{-k(\zeta_2 - s)} \left( I^{\beta-1} |g| \right)(s) ds + \int_0^T \left( I^{\alpha-1} |f| \right)(s) ds \right.$$

$$+ k \int_0^T \int_0^x e^{-k(x-s)} \left( I^{\alpha-1} |f| \right)(s) ds dx \Bigg] + \int_0^{t_2} e^{-k(t_2 - s)} \left( I^{\alpha-1} |f| \right)(s) ds$$

$$+ \int_0^{t_1} e^{-k(t_1 - s)} \left( I^{\alpha-1} |f| \right)(s) ds.$$

$$\leq |\Delta| e^{-k(t_2 - t_1)} \left[ |\mu_2 n_1| \left( I^{\alpha-1} 1 \right) (\eta_2) + k|\mu_2 n_1| \int_0^{\eta_2} e^{-k(\eta_2 - s)} \left( I^{\alpha-1} 1 \right)(s) ds \right.$$

$$+ n_2 \int_0^T \left( I^{\beta-1} 1 \right)(s) ds + kn_2 \int_0^T \int_0^x e^{-k(x-s)} \left( I^{\beta-1} 1 \right)(s) ds dx \Bigg]$$

$$+ |\lambda| e^{-k(t_2 - t_1)} \left[ |\rho_2 n_2| \left( I^{\beta-1} 1 \right) (\zeta_2) + k|\rho_2 n_2| \int_0^{\zeta_2} e^{-k(\zeta_2 - s)} \left( I^{\beta-1} 1 \right)(s) ds \right.$$

$$+ n_1 \int_0^T \left( I^{\alpha-1} 1 \right)(s) ds + kn_1 \int_0^T \int_0^x e^{-k(x-s)} \left( I^{\alpha-1} 1 \right)(s) ds dx \Bigg]$$

$$+ n_1 \left[ \left( \int_0^{t_1} e^{-k(t_2 - s)} - e^{-k(t_1 - s)} \right) \left( I^{\alpha-1} 1 \right)(s) ds + \int_{t_1}^{t_2} e^{-k(t_2 - s)} \left( I^{\alpha-1} 1 \right)(s) ds \right]$$

Hence, we have $\|Z_1(x, y)(t_2) - Z_1(x, y)(t_1)\| \to 0$ independent of x and y as $t_2 \to t_1$. Similarly, $\|Z_2(x, y)(t_2) - Z_2(x, y)(t_1)\| \to 0$ independent of x and y as $t_2 \to t_1$.

Therefore, the operator $Z(x, y)$ is equicontinuous, and thus the operator $Z(x, y)$ is completely continuous.

**Step 2.** Boundedness of operator.

Finally, show that $r = \{(x, y) \in Q \times S : (x, y) = MZ(x, y), N \in [0, 1]\}$ is bounded.

Let:

$$x(t) = MZ_1(x, y)(t), \ y(t) = MZ_2(x, y)(t).$$

Then:

$$|x(t)| \leq M_1(\theta_0 + \theta_1|x| + \theta_2|y|) + N_2(\vartheta_0 + \vartheta_1|x| + \vartheta_2|y|),$$

and:

$$|y(t)| \leq M_3(\theta_0 + \theta_1|x| + \theta_2|y|) + N_4(\vartheta_0 + \vartheta_1|x| + \vartheta_2|y|).$$

So, we get:

$$\|x\| \leq M_1(\theta_0 + \theta_1|x| + \theta_2|y|) + M_2(\vartheta_0 + \vartheta_1|x| + \vartheta_2|y|), \tag{16}$$

and:

$$\|y\| \leq M_3(\theta_0 + \theta_1|x| + \theta_2|y|) + M_4(\vartheta_0 + \vartheta_1|x| + \vartheta_2|y|). \tag{17}$$

From (16) and (17), we obtain:

$$\|x\| + \|y\| \leq (M_1 + M_3)\theta_0 + (M_2 + M_4)\vartheta_0 + ((M_1 + M_3)\theta_1 + (M_2 + M_4)\vartheta_1)\|x\|$$
$$+ ((M_1 + M_3)\theta_2 + (M_2 + M_4)\vartheta_2)\|y\|.$$

Therefore:

$$\|(x,y)\| \leq \frac{(M_1 + M_3)\theta_0 + (M_2 + M_4)\vartheta_0}{M_0},$$

where $M_0 = min\{1 - (M_1 + M_3)\theta_1 - (M_2 + M_4)\vartheta_1, 1 - (M_1 + M_3)\theta_2 - (M_2 + M_4)\vartheta_2\}$ that is $r$ bounded. By the Leray–Schauder theorem, the existence of a solution to the boundary value problem holds true on $[0, T]$. □

## 4. Ulam–Hyers Stability

The Ulam–Hyers stability for our suggested system (1) will be investigated by considering the below inequality:

$$\begin{cases} {}^cD^{\alpha-1}(D+k)x(t) - f(t, x(t), y(t)) \leq \varepsilon_1, \ t \in [0, T], \\ {}^cD^{\beta-1}(D+k)y(t) - g(t, x(t), y(t)) \leq \varepsilon_2, \ t \in [0, T], \end{cases} \tag{18}$$

where $\varepsilon_1, \varepsilon_2$ are given two positive real numbers.

Define the following nonlinear operators $N_1, \ N_2 \ \in C([0, T], \mathbb{R}) \times C([0, T], \mathbb{R}) \rightarrow C([0, T], \mathbb{R})$:

$${}^cD^{\alpha-1}(D+k)x(t) - f(t, x(t), y(t)) = N_1(t), \ t \in [0, T],$$
$${}^cD^{\beta-1}(D+k)y(t) - g(t, x(t), y(t)) = N_2(t), \ t \in [0, T].$$

For some $\varepsilon_1, \varepsilon_2 > 0$, we consider the following inequality:

$$|N_1(t)| \leq \varepsilon, \ |N_2(t)| \leq \varepsilon_2, \ t \in [0, T]. \tag{19}$$

**Definition 4 [27,28].** *The boundary value problem (1) is Ulam–Hyers stable if there exists $M_i, i = 1, 2, 3, 4$ such that for the given $\varepsilon_1, \varepsilon_2 > 0$ and for each solution $(x, y) \in C([0, T] \times \mathbb{R}^2, \mathbb{R})$ of inequality (18), there exists a solution $(x^*, y^*) \in C([0, T] \times \mathbb{R}^2, \mathbb{R})$ of problem (1) with:*

$$\begin{cases} |x(t) - x^*(t)| \leq M_1 \varepsilon_1 + M_2 \varepsilon_2, \ t \in [0, T], \\ |y(t) - y^*(t)| \leq M_3 \varepsilon_1 + M_4 \varepsilon_2, \ t \in [0, T]. \end{cases}$$

**Theorem 3.** *If the assumptions of Theorem 1 hold, then the BVB (1), (2) is Ulam–Hyers stable.*

**Proof.** Let $(x, y) \in C([0, T], \mathbb{R}) \times C([0, T], \mathbb{R})$ be the solution of the problem (1) and (2), satisfying (8) and (9), and let $(x^*, y^*)$ be any solution satisfying:

$$\begin{cases} {}^cD^{\alpha-1}(D+k)x^*(t) = f(t, x^*(t), y^*(t)) + N_1(t), \ t \in [0, T], \\ {}^cD^{\beta-1}(D+k)y^*(t) = g(t, x^*(t), y^*(t)) + N_2(t), \ t \in [0, T]. \end{cases}$$

It follows that:

$$\left| x(t) - \Delta e^{-kt} \left[ \mu_2 (I^{\alpha-1} f)(\eta_2) - k\mu_2 \int_0^{\eta_2} e^{-k(\eta_2-s)} (I^{\alpha-1} f)(s) ds \right. \right.$$

$$- \int_0^T (I^{\beta-1} g)(s) ds + k \int_0^T \int_0^x e^{-k(x-s)} (I^{\beta-1} g)(s) ds dx \bigg]$$

$$- \lambda e^{-kt} \left[ \rho_2 (I^{\beta-1} g)(\zeta_2) - k\rho_2 \int_0^{\zeta_2} e^{-k(\zeta_2-s)} (I^{\beta-1} g)(s) ds - \int_0^T (I^{\alpha-1} f)(s) ds \right.$$

$$+ k \int_0^T \int_0^x e^{-k(x-s)} (I^{\alpha-1} f)(s) ds dx \bigg]$$

$$- \frac{1}{T^2 - \mu_1 \rho_1} \left[ \frac{A\mu_2}{k} (I^{\alpha-1} f)(\eta_2) - A\mu_2 \int_0^{\eta_2} e^{-k(\eta_2-s)} (I^{\alpha-1} f)(s) ds - \frac{A}{k} \int_0^T (I^{\beta-1} g)(s) ds \right.$$

$$+ (A - \rho_1) \int_0^T \int_0^x e^{-k(x-s)} (I^{\beta-1} g)(s) ds dx + \frac{B\rho_2}{k} (I^{\beta-1} g)(\zeta_2)$$

$$- B\rho_2 \int_0^{\zeta_2} e^{-k(\zeta_2-s)} (I^{\beta-1} g)(s) ds - \frac{B}{k} \int_0^T (I^{\alpha-1} f)(s) ds$$

$$+ (B - T) \int_0^T \int_0^x e^{-k(x-s)} (I^{\alpha-1} f)(s) ds dx + T\rho_1 \int_0^{\zeta_1} e^{-k(\zeta_1-s)} (I^{\beta-1} g)(s) ds$$

$$+ \mu_1 \rho_1 \int_0^{\eta_1} e^{-k(\eta_1-s)} (I^{\alpha-1} f)(s) ds \bigg] - \int_0^t e^{-k(t-s)} (I^{\alpha-1} f)(s) ds \bigg|$$

$$\leq |\Delta| e^{-kT} \left[ |\mu_2| (I^{\alpha-1} |N_1(t)|)(\eta_2) + k|\mu_2| \int_0^{\eta_2} e^{-k(\eta_2-s)} (I^{\alpha-1} |N_1(t)|)(s) ds \right.$$

$$+ \int_0^T (I^{\beta-1} |N_2(t)|)(s) ds + k \int_0^T \int_0^x e^{-k(x-s)} (I^{\beta-1} |N_2(t)|)(s) ds dx \bigg]$$

$$+ |\lambda| e^{-kT} \left[ |\rho_2| (I^{\beta-1} |N_2(t)|)(\zeta_2) + k|\rho_2| \int_0^{\zeta_2} e^{-k(\zeta_2-s)} (I^{\beta-1} |N_2(t)|)(s) ds \right.$$

$$+ \int_0^T (I^{\alpha-1} |N_1(t)|)(s) ds + k \int_0^T \int_0^x e^{-k(x-s)} (I^{\alpha-1} |N_1(t)|)(s) ds dx \bigg]$$

$$+ \frac{1}{|\omega|} \left[ \frac{|A\mu_2|}{k} (I^{\alpha-1} |N_1(t)|)(\eta_2) + |A\mu_2| \int_0^{\eta_2} e^{-k(\eta_2-s)} (I^{\alpha-1} |N_1(t)|)(s) ds \right.$$

$$+ \frac{|A|}{k} \int_0^T (I^{\beta-1} |N_2(t)|)(s) ds + |A - \rho_1| \int_0^T \int_0^x e^{-k(x-s)} (I^{\beta-1} |N_2(t)|)(s) ds dx$$

$$+ \frac{|B\rho_2|}{k} (I^{\beta-1} |N_2(t)|)(\zeta_2) + |B\rho_2| \int_0^{\zeta_2} e^{-k(\zeta_2-s)} (I^{\beta-1} |N_2(t)|)(s) ds$$

$$+ \frac{|B|}{k} \int_0^T (I^{\alpha-1} |N_1(t)|)(s) ds + |B - T| \int_0^T \int_0^x e^{-k(x-s)} (I^{\alpha-1} |N_1(t)|)(s) ds dx$$

$$+ T|\rho_1| \int_0^{\zeta_1} e^{-k(\zeta_1-s)} (I^{\beta-1} |N_2(t)|)(s) ds + |\mu_1 \rho_1| \int_0^{\eta_1} e^{-k(\eta_1-s)} (I^{\alpha-1} |N_1(t)|)(s) ds \bigg]$$

$$+ \int_0^T e^{-k(T-s)} (I^{\alpha-1} |N_1(t)|)(s) ds,$$

$$\leq \varepsilon_1 \left[ |\Delta| e^{-kT} \mu_2 \left( (I^{\alpha-1}1)(\eta_2) + k \int_0^{\eta_2} e^{-k(\eta_2-s)} (I^{\alpha-1}1)(s)ds \right) \right.$$

$$+ |\lambda| e^{-kT} \left( \int_0^T (I^{\alpha-1}1)(s)ds + k \int_0^T \int_0^x e^{-k(x-s)} (I^{\alpha-1}1)(s)dsdx \right)$$

$$+ \frac{1}{|\omega|} \left( \frac{|A\mu_2|}{k} (I^{\alpha-1}1)(\eta_2) + |A\mu_2| \int_0^{\eta_2} e^{-k(\eta_2-s)} (I^{\alpha-1}1)(s)ds + \frac{|B|}{k} \int_0^T (I^{\alpha-1}1)(s)ds \right.$$

$$\left. + |B - T| \int_0^T \int_0^x e^{-k(x-s)} (I^{\alpha-1}1)(s)dsdx + |\mu_1\rho_1| \int_0^{\eta_1} e^{-k(\eta_1-s)} (I^{\alpha-1}1)(s)ds \right)$$

$$\left. + \int_0^T e^{-k(T-s)} (I^{\alpha-1}1)(s)ds \right] \tag{20}$$

$$+ \varepsilon_2 \left[ |\Delta| e^{-kT} \left( \int_0^T (I^{\beta-1}1)(s)ds + k \int_0^T \int_0^x e^{-k(x-s)} (I^{\beta-1}1)(s)dsdx \right) \right.$$

$$+ |\lambda| e^{-kT} \left( |\rho_2| (I^{\beta-1}1)(\zeta_2) + k|\rho_2| \int_0^{\zeta_2} e^{-k(\zeta_2-s)} (I^{\beta-1}1)(s)ds \right)$$

$$+ \frac{1}{|\omega|} \left( \frac{|A|}{k} \int_0^T (I^{\beta-1}1)(s)ds + |A - \rho_1| \int_0^T \int_0^x e^{-k(x-s)} (I^{\beta-1}1)(s)dsdx + \frac{|B\rho_2|}{k} (I^{\beta-1}1)(\zeta_2) \right.$$

$$\left. \left. + |B\rho_2| \int_0^{\zeta_2} e^{-k(\zeta_2-s)} (I^{\beta-1}1)(s)ds \right) \right] = \varepsilon_1 M_1 + \varepsilon_2 M_2.$$

By the same method, we can obtain that:

$$|y(t) - y^*(t)| \leq M_3 \varepsilon_1 + M_4 \varepsilon_2, \tag{21}$$

where $M_i, i = 1, 2, 3, 4$ are mentioned before. By (20) and (21), the nonlinear sequential coupled system of Caputo fractional differential equations is Ulam–Hyers stable and consequently, the system (1) is Ulam–Hyers stable. □

**Example 1.** *Consider the following system of a fractional differential equation:*

$$\begin{cases} {}^cD^{1/2}(D+1)x(t) = \dfrac{1}{8\pi\sqrt{49+t^2}} \left( \dfrac{|x(t)|}{1+|x(t)|} + \sin y(t) \right) + \dfrac{1}{4}, \ t \in [0,1] \\[2mm] {}^cD^{1/3}(D+1)y(t) = \dfrac{1}{2\pi(4+t)^2} \left( \sin(x(t)) + \dfrac{y(t)}{1+|x(t)|} \right) + 1, \ t \in [0,1] \\[2mm] \displaystyle\int_0^1 x(s)ds = y(1/2), \int_0^1 x'(s)ds = -2y'(1/2), \\[2mm] \displaystyle\int_0^1 y(s)ds = -3x(1/3), \int_0^1 y'(s)ds = x'(1). \end{cases} \tag{22}$$

Here:

$$k = 1, \alpha = \frac{3}{2}, \beta = \frac{4}{3}, T = 1, \rho_1 = 1, \zeta_1 = \frac{1}{2}, \rho_2 = -2, \zeta_2 = \frac{1}{2}, \mu_1 = -3, \eta_1 = \frac{1}{3}, \mu_2 = 1, \eta_2 = 1.$$

We found:

$$M_1 = 4.5398, M_2 = 4.9766, M_3 = 2.7046, M_4 = 5.872, h_1 = \frac{1}{56\pi}, h_2 = \frac{1}{32\pi}.$$

It's clear that f, g are jointly continuous functions, where:

$$f(t, x, y) = \frac{1}{8\pi\sqrt{49+t^2}} \left( \frac{|x(t)|}{1+|x(t)|} + \sin y(t) \right) + \frac{1}{4},$$

$$g(t, x, y) = \frac{1}{2\pi(4+t)^2} \left( \sin(x(t)) + \frac{y(t)}{1+|x(t)|} \right) + 1.$$

Now, check that $h_1(M_1 + M_3) + h_2(M_2 + M_4) < 1$.
Hence:

$$\frac{1}{56\pi}(7.2444) + \frac{1}{32\pi}(10.8486) = 0.149 < 1.$$

Thus, all the conditions of Theorem 1 are satisfied, and problem (22) has a unique solution on $[0, 1]$.

**Example 2.** *Consider the following system of fractional differential equation:*

$$\begin{cases} {}^cD^{\frac{1}{2}}(D+1)x(t) = \dfrac{1}{60+t^2} + \dfrac{e^{-t}\cos(x(t))}{2\sqrt{6400+t^4}} + \dfrac{|y(t)|}{140(1+x^2(t))} \quad t \in [0,1] \\[2mm] {}^cD^{\frac{1}{3}}(D+1)y(t) = \dfrac{1}{\sqrt{25+t^2}}\cos^2(t) + \dfrac{e^{-t}\sin(x(t))}{130} + \dfrac{y(t)}{120}, \quad t \in [0,1] \\[2mm] \displaystyle\int_0^1 x(s)ds = y(1/2), \int_0^1 x'(s)ds = -2y'(1/2), \\[2mm] \displaystyle\int_0^1 y(s)ds = -3x(1/3), \int_0^1 y'(s)ds = x'(1). \end{cases} \quad (23)$$

Here:

$$k = 1, \alpha = \frac{3}{2}, \beta = \frac{4}{3}, T = 1, \rho_1 = 1, \zeta_1 = \frac{1}{2}, \rho_2 = -2, \zeta_2 = \frac{1}{2}, \mu_1 = -3, \eta_1 = \frac{1}{3}, \mu_2 = 1, \eta_2 = 1.$$

We found:

$$M_1 = 4.5398, M_2 = 4.9766, M_3 = 2.7046, M_4 = 5.872.$$

It's clear that $f, g$ are jointly continuous functions and:

$$|f(t,x,y)| \le \tfrac{1}{60} + \tfrac{1}{160}|x| + \tfrac{1}{140}|y|,$$
$$|g(t,x,y)| \le \tfrac{1}{5} + \tfrac{1}{130}|x| + \tfrac{1}{120}|y|.$$

Thus: $\theta_0 = \tfrac{1}{60}$, $\theta_1 = \tfrac{1}{160}$, $\theta_2 = \tfrac{1}{140}$, $\vartheta_0 = \tfrac{1}{5}$, $\vartheta_1 = \tfrac{1}{130}$, $\vartheta_2 = \tfrac{1}{120}$.
Note that:

$$(M_1 + M_3)\theta_1 + (M_2 + M_4)\vartheta_1 = 0.1288 < 1,$$

and:

$$(M_1 + M_3)\theta_2 + (M_2 + M_4)\vartheta_2 = 0.1422 < 1.$$

Thus, all the conditions of Theorem 2 are satisfied, and problem (23) has at least one solution on $[0, 1]$.

## 5. Conclusions

The existence, stability, and uniqueness for the solution of the coupled system of Caputo-type sequential fractional differential equations that involve integral boundary conditions were investigated. Leray–Schauder's alternative was implemented to show the existence of the proposed system and the Banach's contraction principle was used to examine the uniqueness of the solution. The Ulam–Hyers stability of the proposed system was investigated, and it was found that the presented system was stable and unique; an example has been given to illustrate certain related aspects. The presented approach may be extended to obtain numerical solutions for a coupled system of Caputo-type sequential fractional differential equations, which will be discussed in detail at a later stage.

**Author Contributions:** Writing—original draft, A.A.-k.; Methodology, A.A.-k. software, H.Z.; validation, O.A.; formal analysis, A.A.-k.; investigation, S.B.; writing—original draft preparation, H.Z.; writing—review and editing, A.A.-k., and H.Z.; visualization, O.A. Funding acquisition, S.B. All authors have read and agreed to the published version of the manuscript.

**Funding:** This research received no external funding.

**Institutional Review Board Statement:** Not applicable.

**Informed Consent Statement:** Not applicable.

**Data Availability Statement:** Not applicable.

**Acknowledgments:** The authors wish to thank the anonymous reviewers for their valuable comments and suggestions.

**Conflicts of Interest:** The authors declare no conflict of interest.

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
