# Peer review of "Ulam–Hyers Stability and Uniqueness for Nonlinear Sequential Fractional Differential Equations Involving Integral Boundary Conditions"

_fractalfract, doi:10.3390/fractalfract5040235_

Round 1
Reviewer 1 Report
REVIEWER’s REPORT
On the paper
Ulam–Hyers stability and Uniqueness for Nonlinear Sequential Fractional Differential Equations involving Integral Boundary Conditions
In this paper, the authors investigated the existence, stability, and uniqueness of the solution of the coupled system of Caputo type sequential fractional differential equations. For existence the authors used Leray–Schauder’s alternative theorem and for the uniqueness of the solution the authors used Banach’s contraction principle. Also provided one simple example to validate the given results.
The results are correct and detailed. The methods are very standard one.
The authors should consider the following points while revising this paper:
- In the abstract, the authors need to summarize clearly the problem to be addressed, objectives, methodology, and briefly point out the main results they obtained.
- What is the main contribution and motivation of this paper? The contribution and motivation of the current work should be emphasized in the introduction. Give reasons in detail.
- Include the novelty of the work.
- There are some typos. English should be modified precisely.
- If possible provide one more example.
- The conclusion part should be added with future directions in this field.
- A reference section should be added with some recent works related to this paper and properly cite them.
Considering the above points, I recommend the paper for a MAJOR revision to the journal Fractal and Fractional.

Author Response
the list of response to the reviewer is mentioned in the following attach file

Reviewer 2 Report
The paper is written good and the results are sound. But the paper need.
1. Anstract should be revised.
2. Update the literature with fresh recent work.
3. Improve the presentation.
4. Clarify conclusion.
5.Rest of paper is ok
Author Response
Please see the attach file

Reviewer 3 Report
The referee strongly recommends this excellent paper for publication in Fractal and Fractional after necessary/minor changes; please see the attachment.

Author Response
At the first, the authors would thank the reviewer for his valuable time in reviewing the paper and most of the recommended corrections have been corrected.
Round 2
Reviewer 1 Report
Dear Editor
The authors carried out all the corrections suggested by me. Hence I recommend this paper to publish in your esteemed journal.